# Effects of Different Durations at Fixed Intensity Exercise on Internal Load and Recovery—A Feasibility Pilot Study on Duration as an Independent Variable for Exercise Prescription

**DOI:** 10.3390/jfmk7030054

**Published:** 2022-07-21

**Authors:** Philipp Birnbaumer, Lena Weiner, Tanja Handl, Gerhard Tschakert, Peter Hofmann

**Affiliations:** Exercise Physiology, Training & Training Therapy Research Group, Institute of Human Movement Science, Sport & Health, University of Graz, 8010 Graz, Austria; philipp.birnbaumer@uni-graz.at (P.B.); l.weiner@gmx.at (L.W.); tanja.handl@koeroesi.at (T.H.); gerhard.tschakert@uni-graz.at (G.T.)

**Keywords:** training, duration thresholds, recovery, adaption, intensity, exercise prescription

## Abstract

Duration is a rarely investigated marker of exercise prescription. The aim of this study was to test the feasibility of the methodological approach, assessing effects of different duration constant-load exercise (CLE) on physiological responses (internal load) and recovery kinetics. Seven subjects performed an incremental exercise (IE) test, one maximal duration CLE at 77.6 ± 4.8% $\dot{V}O_{2max}$, and CLE’s at 20%, 40%, and 70% of maximum duration. Heart rate (HR), blood lactate (La), and glucose (Glu) concentrations were measured. Before, 4, 24, and 48 h after CLE’s, submaximal IE tests were performed. HR variability (HRV) was assessed in orthostatic tests (OT). Rating of perceived exertion (RPE) was obtained during all tests. CLE’s were performed at 182 ± 27 W. HR_peak_, La_peak_, $\dot{V}E_{\mathrm{peak}}$, and RPE_peak_ were significantly higher in CLE’s with longer duration. No significant differences were found between CLE’s for recovery kinetics for HR, La, and Glu in the submaximal IE and for HRV or OT. Despite no significant differences, recovery kinetics were found as expected, indicating the feasibility of the applied methods. Maximum tests and recovery tests closer to CLE’s termination are suggested to better display recovery kinetics. These findings are a first step to prescription of exercise by both intensity and duration on an individual basis.

## 1. Introduction 

It is well accepted that individual exercise prescription is an important tool for regulating exercise training in health and disease; however, the usual models primarily focus on exercise intensity and easy-to-apply methods for intensity prescription which have been critically discussed recently [1]. Apart from intensity, additional variables need to be respected and summarized in the so-called F.I.T.T. principle, where training workload is prescribed by frequency, intensity, and time (duration/volume) as well as type of exercise [2]. Currently, the main variable of individual exercise prescription is almost exclusively intensity, neglecting the effects of duration on fatigue, recovery kinetics, and adaptation [3]. The importance of duration in exercise prescription is supported by studies showing that, independent of intensity, frequency and length of interventions and the improvement of maximum oxygen uptake ($\dot{V}O_{2max}$) was dependent on session duration [4]. It may therefore be proposed that exercise duration also needs to be prescribed individually, to set a physiologically well-grounded basis to specifically ensure a valid comparison of scientific studies [5]. We recently presented a theoretical basis to individualize exercise duration, recognizing a major lack of experimental data around exercise duration effects [6].

Overall, there is a lack of studies on exercise duration effects which has been critically highlighted by Tremblay et al. [7]. Some duration effects have been summarized by Tschakert et al. [8] for muscular micro-RNA response, hormone dynamics, hemodynamics, heart rate (HR), heart rate variability (HRV), and post-exercise oxygen consumption, as well as metabolism and immune response. However, no individualized approach regarding the prescription of duration has been applied rather than comparing fixed durations. Therefore, detailed individual duration-dependent differences for acute effects as well as chronic adaptations are still missing. Currently, studies prescribe exercise duration by means of fixed absolute values, such as 15, 30, 45, 60, 90, or 120 min, but hardly on an individual basis (e.g., as a percentage of maximal duration at a given intensity) [8,9]. This is in contrast to the individualized prescription of intensity, applying well-accepted thresholds such as the first and second turn points for lactate (LTP_1_, LTP_2_) or ventilation (VT_1_, VT_2_) [10,11]. Regarding duration, it is obvious from the literature that we still have no relevant information regarding individual markers such as maximum duration (t_max_) and derived duration domains. Currently, models for an individualized prescription of exercise duration are extremely rare and rather theoretical [6]. In a first attempt we could show in a small group of trained subjects that for most if not all physiological variables, defined duration thresholds can be determined objectively which were significantly related to t_max_ [8]. Similar results have been shown already by Mezzani et al. [11,12] in trained and untrained subjects as well as patients where t_max_ was dependent on performance. Interestingly, percentages of t_max_ at these duration thresholds were similar across subjects and comparable to already published markers in a theoretical concept [13]. It may be therefore proposed that each individual has a certain capacity [14] (p. 495) to sustain a comparable relative workload (e.g., percentage of threshold performance) which needs to be respected in exercise prescription. The closer one gets to t_max_, the greater the grade of homeostatic disturbance and fatigue will become, consequently altering recovery kinetics [15,16,17,18] as well as subsequent adaptation [6,7,13,19,20]. In his theoretical position, Platonov [13] proposed that increasing duration for a given intensity leads to an increase in fatigue, reduces the sustainable intensity when exceeding t_max_, and consequently terminates exercise, inducing prolonged recovery times, and subsequently increased adaptation with increased performance. Reducing the duration to a certain sub-maximal percentage of t_max_ is suggested to induce a so-called compensated fatigue without a loss in performance but higher efforts to sustain exercise and, consequently, shorter recovery time as well as less adaptations compared to a t_max_ approach. Reducing duration further to 40–60% of the maximal duration is suggested to induce only a small amount of fatigue, if any, and, therefore, no acute performance-increasing effect is expected, but rather stabilizing of the actual performance level. A duration of less than 20% of t_max_ is suggested to induce only a functional stimulation adequate for regeneration. So, for one chosen intensity, four duration domains have been proposed [6,13] which are suggested to induce a different amount of fatigue characterized by performance, recovery kinetics, and adaptation. In training practice this would mean that independent from intensity training sessions with a regenerative, stabilizing or developing character could be realized by prescribing duration relative to t_max_. Furthermore, individual prescription of duration would allow a more accurate estimation of the expected fatigue in a complex training situation. Currently, no experimental data or methodological approaches to the problem are available to support these assumptions. Another possibility to monitor and prescribe training is the so-called training load concept which combines intensity and duration to determine the cumulative amount of stress of a single or multiple training session. However, authors questioned the validity of this concept because the present use of session duration was suggested to confound the current training load metrics [21]. All concepts rely on the assumption that training load determines fatigue and duration of recovery processes although applying different physiological markers for determination [22].

Several methods such as HR and HRV are well-known variables to describe fatigue and recovery kinetics. However, fatigue during exercise is a very complex concept and processes are dependent on exercise intensity [23]. As the internal load determines the training outcome [22], monitoring of, e.g., HR or La represents an easy and practical option to differentiate between strain of different exercise types. Furthermore, the use of HRV measures to examine training load, disturbance of body’s homeostasis, and recovery state after training has been well examined and was shown to be eligible to determine acute response to exercise and long term training effects [24,25,26,27,28]. HRV is reduced beyond the pre-level during exercise and acute recovery, which was shown to be influenced by the type, intensity, and duration of exercise [29]. After sufficient recovery, a rebound of HRV can be observed which is usually associated with improved performance [30]. However, regarding different durations at the same intensity no data on the feasibility of HRV measures and the internal load are yet available.

To answer these pending questions, we conducted a first small proof-of-concept study to investigate the feasibility of the methodological approach to assess internal load, recovery kinetics, and short-term adaptations of four constant-load cycle ergometer exercise tests (CLE) at the same individual intensity but different individual duration expressed as percentages of t_max_. The aim of this study was therefore to test if the chosen methodological approach allows determination of changes in physiological measures during and after constant-load exercise and determination of different effects for heart rate, heart rate variability, lactate, and glucose concentration and performance in sub-maximal incremental exercise tests as well as changes in orthostatic tests conducted pre and post CLE tests of different duration. Our main hypothesis was that the chosen methodological approach is feasible to prescribe differences in the internal load and recovery kinetics with respect to duration at the same exercise intensity.

## 2. Materials and Methods

### 2.1. Subject Characteristics

Seven healthy, young, moderately trained male (m) (5) and female (f) (2) subjects (age: 27.2 ± 2.6 years; body mass index (BMI): 24.1 ± 1.9 kg/m^2^; height: 1.73 ± 0.09 m) as part of another study [8] participated in this study. All subjects were highly physical active, participating in various sports classes within their leisure time or studies and performing different sports on a regular basis (2–3 times per week) but not in a competitive way. All subjects were familiar with intensive cycling exercise, but only three performed regular cycling exercise. Only active male and female adults (20–38 years) were allowed to participate and were included in the study after passing a medical checkup. All subjects signed a written informed consent, and the study design was approved by the Ethics Committee of the University of Graz (39/72/63 ex 2017/18).

### 2.2. Experimental Design

All subjects performed an incremental cycle ergometer exercise (IE) up to exhaustion to determine exercise intensity for CLE as well as $\dot{V}O_{2max}$, the maximum power output (P_max_) and the power output at LTP_1_ and LTP_2_ as well as VT_1_ and VT_2_. After a 3 min rest and a 3 min warm-up phase at 20 W (f) or 40 W (m), intensity was increased by 15 W (f) or 20 W (m) per minute up to exhaustion, followed by 3 min of active recovery (same intensity as in warm-up) and 3 min of passive recovery. At least one week after the IE test, all subjects started with the all-out duration (CLE100) tests. Workload for CLE was set at an exercise intensity of 10% P_max_ below the power output at LTP_2_ (PLTP_2_) from maximum IE. Subjects were encouraged to sustain the target workload as long as possible and were supported by strong verbal encouragement. Then, duration was shortened to 70% (CLE70), 40% (CLE40) and 20% (CLE20) of CLE100 according to the concept of Platonov [13]. These tests were performed between 8 and 11 AM in a randomized order with at least one week of break in-between. All CLE tests started with a 3 min resting phase to obtain pre-exercise measures. At the start of exercise, intensity was increased immediately to the target workload and kept constant for the pre-determined individual duration as a percentage of CE100, followed by a 3 min active (f: 20 W or m: 40 W) and a 3 min passive recovery after termination of the workout phase. No food or fluid intake was allowed during all tests. Immediately before (baseline) as well as at 4, 24, and 48 h of recovery (Rec 4, Rec 24, and Rec 48) after each CLE, subjects performed sub-maximal IE tests up to a peak intensity slightly (15–20 W) above the pre-determined LTP_2_. Additionally, subjects performed an orthostatic test (OT) [31] to determine HR and HRV parameters at baseline and before each submaximal IE during the recovery phase at Rec4, Rec24, and Rec48. The test consisted of five minutes lying in resting supine position followed by three minutes of standing quietly. Subjects lay down for 5 to 10 min before the test started and were told by the investigator to stand up immediately after the 5 min supine rest phase. During the whole test, subjects were in a quiet place without any disturbance, and they were instructed to remain quiet and motionless unless standing up. All measures were collected under spontaneous breathing conditions.

Subjects were told to refrain from any strenuous activity 24 h before the maximum IE, the CLE tests as well as between the recovery tests. There were no specific restrictions with regard to the nutritional intake before and between the tests, however subjects were told to have their standard meals, taken 1 to 2 h in advance of the tests.

### 2.3. Measurements and Analysis

All tests were performed on an electronically braked cycle ergometer (Monark Ergomedic 839 E, Monark, Vansbro, Sweden) in a standard laboratory with defined climate conditions set at 21 °C. HR (Polar S810i, Polar Electro, Kempele, Finland) and ventilatory parameters (ZAN 600, ZAN, Steyr-Dietach, Austria) were continuously measured during all tests, except for the baseline tests directly before CLE, where no ventilatory parameters were measured. In addition to air conditioning, a fan was used for cooling during the CLE tests.

Subjects’ height was measured once before the maximum IE with a stadiometer and their weight was measured before every CLE and submaximal IE with a commercial electronic scale (Soehnle Slim Design Silver, Soehnle Industrial Solutions, Backnang, Germany).

HRV data were recorded via a Bluetooth HR sensor (H10, Polar Electro, Kempele Finland) paired with a Polar watch (Polar S810i, Polar Electro, Kempele, Finland). From R–R interval files (measured at 1000 Hz), the root mean square of successive R–R interval differences (RMSSD) and the standard deviation of Poincaré plot perpendicular to the line-of-identity (SD1) and along the line of identity (SD2) were determined with the software Polar Pro Trainer 5 (Polar Electro, Finland). Data were analyzed within the second and forth minute for the lying position and within the last two minutes for the standing position. HRV raw data were checked for errors and standard corrections as given by the software were applied, when necessary, in some rare cases. The applied HR measures have been proven valid [32].

Blood lactate (La) and glucose (Glu) concentrations were obtained from capillary blood samples taken from the hyperemized ear lobe during all exercise tests. In the IE test, blood samples were taken during the resting phase and at the end of the warm-up phase, at the end of each workload step as well as at the end of active and passive recovery. During the CE tests, blood samples were taken during the resting period, after 2, 4, 6, 8, 10, 15, 20, 25, 35, 45, 55, 65 min, etc. until t_max_ was reached, as well as at the end of active and passive recovery. Blood samples were used for the determination of La and Glu by means of a fully enzymatic amperometric method (Biosen S-line, EKF-Diagnostics, Barleben, Germany). At the same time point blood samples were taken, subjects reported their rating of perceived exertion (RPE) according to the Borg Scale [33].

From the maximum IE, LTP_1_ and LTP_2_ as well as VT_1_ and VT_2_ were assessed by means of a computer-aided linear regression break point method (ProSport, Graz, Austria) [34]. Regions of interest were defined for the determination of each turn point. The first threshold was exclusively determined between the first workload and 66% P_max_, and the second turn point between the first turn point and P_max_. VT_1_ and VT_2_ were determined within the same regions of interest. VT_1_ was defined as the first increase in ventilation ($\dot{V}E$) accompanied by an increase in the equivalent for oxygen uptake ($\dot{V}E$/$\dot{V}O_{2}$) without an increase in the equivalent for carbon dioxide output ($\dot{V}E$/$\dot{V}CO_{2}$). VT_2_ was defined as the second increase in $\dot{V}E$ accompanied by an increase in both $\dot{V}E$/$\dot{V}O_{2}$ and $\dot{V}E$/$\dot{V}CO_{2}$. From sub-maximal IE only LTP_1_ was determined for baseline and recovery tests applying the same method. VT_1_ was determined only for the 4, 24, and 48 h recovery tests.

Differences in recovery kinetics between exercise with the same individual intensity but different individual durations were characterized by variables of the submaximal IE-tests (Rec 4, Rec 24, and Rec 48). HR, La, and Glu values measured at rest, LTP_1_, and peak as well as the power output at LTP_1_ were compared to the baseline IE test. For HR, higher values compared to baseline values are interpreted as a loss of performance. For power output at LTP_1_, La, and Glu, relative lower values are interpreted as substrate depletion and a higher grade of homeostatic disturbance.

### 2.4. Statistics

All data are presented as means ± standard deviation (SD) and were tested for normal distribution by the Shapiro–Wilk test. For statistical analysis, GraphPad Prism 8.0.2 (GraphPad Software, San Diego, CA, USA) was used. One-way repeated-measures ANOVA was used to investigate the effect of a different duration at constant intensity on peak HR, $\dot{V}O_{2}$, La, and RPE in CLE. To investigate the effect of CLE duration on fatigue, recovery, and adaption, HR, La, Glu, and HRV variables from submaximal incremental tests (Rec4, Rec24, and Rec 48) were expressed as percentage changes of the values in the pre-exercise tests (baseline). A 4 × 3 ANOVA with two within factors was used to investigate the main effects of CLE duration and recovery time, as well as the interaction between CLE duration and recovery time. When a significant effect was found, the Tukey post-hoc test was performed for normal distributed variables and a Dunn’s multiple comparison test if variables were not normally distributed. Additional, effect size was calculated as Cohen’s d (d) [35]. For all tests, a *p* value < 0.05 was considered significant.

## 3. Results

Maximal as well as sub-maximal values from the maximum IE are presented in Table 1. Relative $\dot{V}O_{2max}$ was found at 52.88 ± 5.11 mL/kg/min. Power output determined at LTP_1_ and LTP_2_ was not significantly different compared to the power output at VT_1_ and VT_2_. CLE intensity was set at 10% P_max_ below PLTP_2_, which was 182 ± 27 W equivalent to 77.6 ± 4.8% $\dot{V}O_{2max}$. The CLE100 test was accomplished at a mean maximum duration of 75 ± 10 min. Duration for the subsequent submaximal duration CLE tests was therefore set at 53 ± 7 min in CLE70, 30 ± 4 min in CLE40, and 15 ± 2 min in CLE20.

### 3.1. Peak Values during CLE

Peak values for HR, La, $\dot{V}E$, and RPE at termination were significantly different between CLE tests, but not for $\dot{V}O_{2}$. Mean HR_peak_ at CLE100 (177 ± 8 bpm) was significantly higher compared to HR_peak_ in CLE40 (161 ± 11 bpm; *p* = 0.003; d = 0.63) and CLE20 (154 ± 11 bpm; *p* < 0.001, d = 0.87). Except for the comparison of HR_peak_ in CLE70 (172 ± 13 bpm) and CLE20 (*p* = 0.002, d = 0.78), no significant differences were found between the other tests. Dunn’s multiple comparison test for not normally distributed variables revealed significantly higher La_peak_ in CLE100 (5.26 ± 1.8 mmol/L) compared to CLE20 (3.38 ± 1.46 mmol/L; *p* = 0.019, d = 0.48) but not CLE70 (4.75 ± 1.70 mmol/L) and CLE40 (4.52 ± 1.99 mmol/L). Significant differences were found for $\dot{V}E_{\mathrm{peak}}$ in CLE100 (94.17 ± 12.98 L/min) and CLE70 (88.82 ± 17.15 L/min) compared to CLE20 (70.85 ± 10.94 L/min; *p* = 0.002, d = 0.71/0.016, d = 0.59) but no significant differences were found between the other tests (CLE40 (80.94 ± 19.99 L/min)). Similarly, we found peak RPE values significantly higher in CLE100 (19.57 ± 1.05) compared to CLE40 (13.29 ± 1.03; *p* = 0.011, d = 0.92) and CLE20 (12.71 ± 0.70; *p* = 0.002, d = 0.93), but not CLE70 (16.71 ± 2.49). Figure 1 shows HR and La (A) as well as $\dot{V}O_{2}$ and $\dot{V}E$ (B) kinetics during the CLE tests with different duration. With increasing duration, HR, La, and $\dot{V}E$ increased continuously up to a peak value at CLE100 but did not reach maximal values according to the maximum IE test. However, peak $\dot{V}O_{2}$ remained constant, independent of the duration of the test equivalent to the constant workload applied. Peak values remained close to second threshold values from the maximum IE for CLE20 and CLE40 but were significantly elevated above second threshold values in CLE100 for HR, La, and $\dot{V}E$, and only significant for $\dot{V}E$ in CLE70.

### 3.2. Loss of Performance during CLE

As one possibility to estimate the degree of fatigue for CLE70 and CLE100 without changes in external load, we suggested a calculated “virtual” performance loss. The difference in $\dot{V}E_{\mathrm{peak}}$ gives an equivalent for power determined in the maximum IE test which had to be compensated by means of an increase in $\dot{V}E$ (30 s mean value) between 35 min of CLE and peak values at the end of CLE70 and CLE100 exercise. This calculated mean performance loss was found significantly different (*p* = 0.018) at 14.46 ± 14.62 W in CLE70 compared to 40.23 ± 15.71 W in CLE100. Per definition, no loss in performance was determined for CLE20 and CLE40. For example, VE difference between minute 35 of CLE100 and peak was 19.7 L/min at the same workload of 211 W. $\dot{V}E$ in the incremental test at this workload was 65.2 L/min. Adding the difference of 19.7 L/min results in a $\dot{V}E$ of 84.9 L/min, which corresponds to 259 W in the maximum IE test, meaning a “virtual” performance loss for this subject of 48 W.

### 3.3. Recovery after CLE Compared to Baseline Values

Two-way repeated measures ANOVAs revealed no statistically significant main or interaction effects for the variables HR, La, and Glu measured at rest, LTP_1_, and peak of the submaximal IE tests. Figure 2 shows the values 4, 24, and 48 h after CLE as a percentage of baseline values obtained before each CLE. Mean HR values at rest and LTP_1_ showed increased values in Rec4 in all tests except CLE20. In Rec24 and Rec48 values returned to baseline. HR_peak_ values are comparable to baseline and showed less variation depending on duration and Rec tests. Regarding La, peak values of all recovery tests showed a consistent pattern, where La concentration was the lowest in Rec4 and increased in Rec24 and Rec48. La concentration as a percentage of baseline measures was lower after CLE tests with longer durations. La values at rest and LTP_1_ were lower in all tests compared to baseline, except for the CLE70 at Rec4 and Rec48. Glu values at rest were higher in Rec4 after all CLE tests and returned to baseline in Rec24 and Rec48. Contrarily, Glu values at peak were lower in Rec4 compared to baseline values and increased above baseline in CLE100 and CLE70 in Rec24 before heading back towards baseline in Rec48. This was not seen in CLE40 and CLE20. Comparable measures can be seen for Glu at LTP_1_. Performance determined at LTP_1_ was not statistically different between tests and was comparable for all CLE tests at baseline Rec4, Rec24, and Rec48 (Table 2).

### 3.4. HRV in Orthostatic Tests

No statistically significant effects of CLE duration and recovery time were found for HRV variables RMSSD, SD1, and SD2 in lying as well as in standing position before and after 4, 24, and 48 h of CLE exercise. However, HRV in lying position showed substantially decreased values in CLE100 and CLE70 at Rec4. However, values increased above baseline in Rec24 and were still higher compared to baseline in Rec48. In CLE40 and CLE20, HRV values were not decreased in Rec4 and were higher or comparable to baseline in all tests. Standing mean HRV values were comparable to baseline values in all four duration tests at Rec4, but values were substantially higher in CLE100 and CLE70 at Rec24 and Rec48 (Figure 3).

## 4. Discussion

Our methodological approach to assessing internal load, recovery kinetics, and short-term adaptation induced by constant-load cycle ergometer exercise at one specific intensity, but different durations, was successful in principle, showing significant time effects for various physiological variables. However, no statistical differences for recovery kinetics were found due to the small sample size. Comparison of internal load between CLE’s showed greater effects between CLE100 and CLE20, but no differences were found between CLE70 and CLE20. However, HR_peak_, $\dot{V}E_{\mathrm{peak}}$, and La_peak_ increased with duration, indicating a duration dependent non-linear increase in fatigue during CLE. The calculation of a virtual loss of performance showed a clear and significant duration effect for CLE70 and CLE100 but not for CLE20 and CLE40, indicating a duration-dependent non-linear increase in fatigue during CLE. Sub-maximal incremental cycle ergometer as well as orthostatic stress tests conducted before and 4, 24, and 48 h after CLE presented the expected but not significant changes, with stronger changes in longer duration exercise. Despite the limits of such a small sample size, our novel approach to prescribe exercise duration based on fixed percentages of the individual maximal duration revealed significant duration dependent differences in physiological measures during CE, such as proposed by Platonov [13] (p. 40). Peak values of HR, La, $\dot{V}E$, and RPE were significantly higher in CLE100 compared to the submaximal durations CLE20 and CLE40, but not CLE70. Effect size calculations revealed a large effect for RPE (d ≥ 8), a medium effect for HR and VE (d ≥ 0.5), and a small effect for La (d ≥ 0.2). The lack of difference between CLE100 and CLE70 can be explained by the fact that both durations are within the same phase of compensated fatigue as shown recently [8]. No differences between CLE’s were detected for oxygen uptake and a clear steady state was found, which implies maintenance of efficiency in all CLE tests [36,37]. However, all other variables presented a significantly higher degree of internal load in tests with a longer duration.

The applied fixed durations of 40 and 70% CLE100 according to Platonov [13] are close to individual duration thresholds detected at 35.4 ± 2.7% and 67.9 ± 2.4% of CLE100, as has been recently presented by our working group [8]. However, CLE20 duration was longer than the first duration threshold detected at 3.7 ± 1.2% CLE100. The prescription of exercise duration applied in our study was according to the four-phase concept of Platonov [13] as modified by our working group [6,8]. As shown recently, constant intensity exercise up to termination defines the end of phase 3 within this concept. Phase 4 is characterized by developing clear fatigue and a distinct reduction in performance which has not been investigated here. It is expected that further increasing duration upon CLE100 by reducing intensity in small steps will additionally increase recovery time but also adaptation [38]. To standardize workload, we did not include such a phase 4 in our investigation, however, it may be necessary in future studies. As the aim of the study was to prove the concept of different duration effects on internal load and recovery kinetics, we did not determine duration thresholds in CLE100 in this part of the study, however, these thresholds were shown recently [8].

Regarding possible effects of duration on recovery and adaption kinetics, we performed submaximal IE as well as orthostatic stress tests as a baseline immediately before the CLE tests and three recovery tests at 4, 24, and 48 h after CLE. We expected significant changes in performance and physiological measures to be greater in CLE tests with longer duration compared to shorter tests. With increasing recovery time, these parameters should return to their respective baselines reaching a higher level, according to the supercompensation concept [38]. Such effects can be seen in our data, but the differences between tests are not statistically significant due to the sample size and the fact that greater changes may be developed only at maximal exercise. The applied approach, however, may be successful in determining individual recovery kinetics and adaptation processes in larger sufficiently powered studies. A post-hoc power analysis revealed approximately 100 subjects to prove significant effects for highly variable measures such as HRV.

Critically, it needs to be questioned if 4 h of rest is too long for analyzing fatigue after different duration exercises, masked by rather fast recovery within a short period of time (within one hour or even faster) [39,40,41]. Exercise tests directly after CLE, however, are rather demanding and motivation/activation potential of participants may be a severe limit for such an approach. From our perspective, the first recovery test should be closer to CLE termination and within one hour of recovery, a strategy which may be limited to subjects which are experienced in exercise training. Additionally, performance at LTP_1_ in submaximal CLE tests presented no significant changes and was found independent of duration and recovery status, although lactate concentration varied, indicating differences in glycogen depletion [42]. Other variables such as ventilation derived thresholds (VT_1_) and especially VT_2_ as well as maximum performance and related variables from a maximum incremental exercise test are therefore suggested as better measures. A promising method appears to determine the acute performance decrement (APD), recently presented by Passfield et al. [21]. With this method, time to task failure or time-trial performance are used to determine immediate effects on subsequent performance. Time to task failure was characterized as a very sensitive method and was shown to even detect a change of 10 W in a 30 min session. Therefore, we suggest that maximal constant-load tests according to the APD concept [9,21], or maximum incremental exercise tests should be performed to investigate recovery kinetics of performance. On the other hand, maximum tests directly after fatiguing exercise as well as after 24 and 48 h (or even more frequently) may be difficult to perform due to lack of motivation of subjects as well as safety reasons. Additionally, such a number of tests is suggested to delay recovery processes, influencing outcome and interpretation. For that reason, submaximal tests were applied in our study, however also with some limitations for the interpretation of recovery processes.

As an additional limit of the study, we need to mention that we just investigated one single intensity below the second threshold at approx. 78% of $\dot{V}O_{2max}$. Fatigue quality, severity and development are clearly dependent on the metabolic domain (below or above first and second threshold), type of exercise (constant-load, interval, sprint type) and individual variables such as age, sex, and cardio-respiratory fitness [40]. In further studies, specific details such as training status, sex, and sports specificity need to be respected. Furthermore, we did not standardize diet or control for additional exercise during the 48 h after CLE. Although subjects were told to perform no additional exercise and maintain their normal eating habits, differences in their activity levels and food intake might have influenced recovery kinetics. As all these variables will have an influence, our approach can just act as a feasibility study to set the methodological basis for follow-up studies to investigate these pending questions. Additionally physical activity and nutrition need to be specifically controlled during the recovery phase in future fully powered studies.

To evaluate differences in recovery kinetics we used HRV measures such as RMSSD, SD1, and SD2 to quantify the individual autonomic response to CE exercise with different durations. Due to the high sympathetic drive, HRV is clearly reduced beyond the pre-level during exercise but returns to baseline or even higher levels after sufficient recovery [30]. In our study we could not show any statistically significant effect of duration on HRV measures as the sample size was too small for such an analysis. However, HRV measures in lying position presented a clear rebound effect both in CLE100 and CLE70. Exercise at the same intensity for a significantly shorter duration in CLE40 and CLE20 did not cause such effects (Figure 3). Previous studies have shown such a rebound effect beyond the pre-training level during subsequent rest or a lighter training period for acute aerobic exercise and intense training periods which initially decrease cardiac vagal outflow [30,43,44,45,46]. The rebound of HRV was associated with improved performance in athletes [30]. Based on the measurement of RMSSD, SD1, and SD2 we therefore suggest that maximum (CLE100) or close to maximum (CLE70) duration exercise causes stronger disturbances of homeostasis followed by an increased vagal modulation after 48 h of recovery. Subsequently, higher adaptation effects are expected. Interestingly, CLE40 and CLE20 HRV measures were already increased above baseline after four hours of recovery and remained close to baseline or slightly higher at 24 and 48 h. However, no severe disturbance of homeostasis was detected after CE20 and CE40 at the chosen moderate-to-high intensity at about 78% $\dot{V}O_{2max}$, which seems to have a recovery effect, as proposed by Platonov [13] (p. 51). Seiler et al. [25] have already demonstrated that 30 min of constant-load exercise at the second threshold or intermittent exercise above the second threshold induced a significant delay of HRV recovery. They showed a return to baseline of HRV measures approximately 30 min after the exercise sessions, which was faster in highly trained male endurance athletes compared to less trained subjects. Exercise duration of 30 min in CLE70 in our study is comparable, however, HRV values were still reduced after 4 h of recovery, although exercise intensity was 10% of P_max_, below the second threshold. It can be assumed that highly trained athletes can tolerate longer exercise durations [25] at such a threshold intensity, although the relation to the maximum duration is missing in this study group. Similar effects were shown by Tschakert et al. [8] for a comparable study group with a substantial interindividual variability in t_max_, also comparable to results presented by Mezzani et al. [12,47] for healthy subjects and patients. A study by Myllymäki et al. [27] investigated the influence of exercise corresponding to 60% $\dot{V}O_{2max}$ and varying durations of 30, 60, and 90 min on nocturnal HRV. These authors detected delayed recovery with increased exercise duration. Similar was shown by Hynynen et al. [24], demonstrating that running a marathon distance (217 min) reduced HRV values significantly longer compared to moderate endurance exercise sessions with a duration of about 52 min. In support of these results, longer duration CLE delayed the return to baseline HRV values in our study.

Measuring HRV variables as late as 4 h after CLE exercise caused only small deviations, if any, compared to baseline measures in lying position and none in standing position. Comparable to performance measures, recovery measures need to be as close as possible to exercise termination to quantify fatigue after demanding exercise and should be continued up to 48 h to highlight recovery kinetics in detail. To quantify the time course, a greater number than the applied three recovery tests are necessary. With respect to Platonov’s [13] (p. 48) assumptions of a three-phase recovery kinetic after a single strenuous exercise bout, measures directly after termination of exercise as well as after 30 min, 1 h, 3–6 h, and after 12, 24, 36 and 48 h seem necessary to detect full recovery and the start of the super-compensation phase. Buchheit et al. [48] showed that increases in maximal aerobic running speed were correlated to increases in resting HRV.

Additional benefit in assessing fatigue may be derived from RPE ratings, which has been shown recently by Pind et al. [49] when investigating RPE during a four-week low intensity high volume period in rowers. These authors found significant differences for fatigue at the end of the training block, although relative intensity and duration as well as volume were the same. They concluded that manipulating exercise duration at a similar training intensity results in a different level of fatigue, which is supported by our results showing an increase of RPE from about 12 in CLE20 to about 19 in CLE100. Session RPE may be a useful and easy to apply additional marker to quantify fatigue [50].

Regarding the F.I.T.T. principle [2], frequency of exercise sessions is a dependent variable of fatigue and recovery needs, induced by a specific combination of intensity and duration such as is shown in this study for just one specific type and intensity of exercise. It is therefore obvious that additional studies, including different intensities and types of exercise in various groups of subjects different in age, sex, and cardio-respiratory fitness, are necessary to complete the understanding of duration of exercise as an independent variable in exercise prescription.

## 5. Conclusions

The time course of changes of peak values of HR, La, Glu, and $\dot{V}E$ supports our theoretical approach, showing greater changes of these variables with longer CLE at the same intensity. The results indicate an independent relevance of the marker “duration” with regard to internal load, recovery time, and adaptation for constant-load endurance type exercise between the first and second ventilatory thresholds. These findings are a first step to build up a framework for prescribing exercise by intensity and duration on an individual basis. However, the concept needs to be further developed and applied for the whole spectrum of intensities, methods, and types of exercise.

## Figures and Tables

**Figure 1 jfmk-07-00054-f001:**
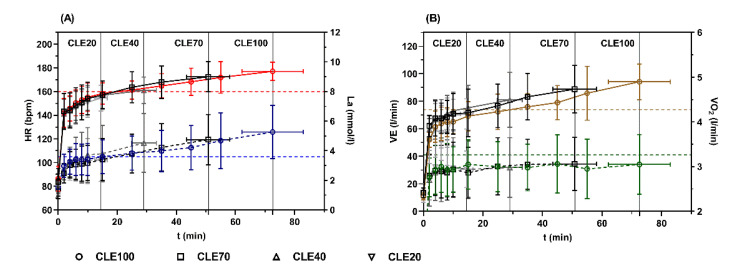
(**A**) Heart rate (HR) (solid lines) and lactate concentration (La) (dashed lines) as well as (**B**) ventilation ($\dot{V}E$) (solid lines) and oxygen uptake ($\dot{V}O_{2}$ ) (dashed lines) during constant-load cycle ergometer exercise tests (CLE) with maximum duration (CLE100) and 70, 40, and 20% (CLE70, CLE40 and CLE20) of CLE100 at the same intensity determined from a maximal incremental cycle ergometer exercise test (10% of P_max_ below PLTP_2_). Vertical dotted lines mark the peak duration of the single CE tests. Dashed horizontal lines indicate the mean values for HR (red), La (blue), $\dot{V}E$ (brown), and $\dot{V}O_{2}$ (green) at the second threshold from the maximum incremental cycle ergometer test.

**Figure 2 jfmk-07-00054-f002:**
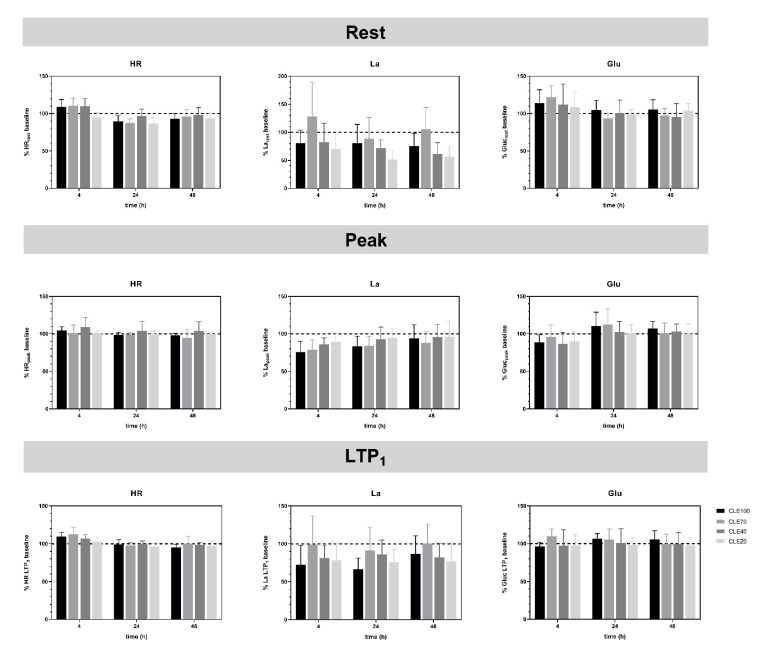
Heart rate (HR), lactate (La), and glucose (Glu) values at rest, LTP_1_, and peak from submaximal incremental ergometer exercise test (IE) performed at 4, 24, and 48 h (Rec4, Rec24, and Rec 48) after constant-load cycle ergometer exercise tests with maximum duration (CLE100) and 70, 40, and 20% (CLE70, CLE40, and CLE20) of CLE100 at the same intensity, shown as a percentage of the baseline IE values determined immediately before each CLE.

**Figure 3 jfmk-07-00054-f003:**
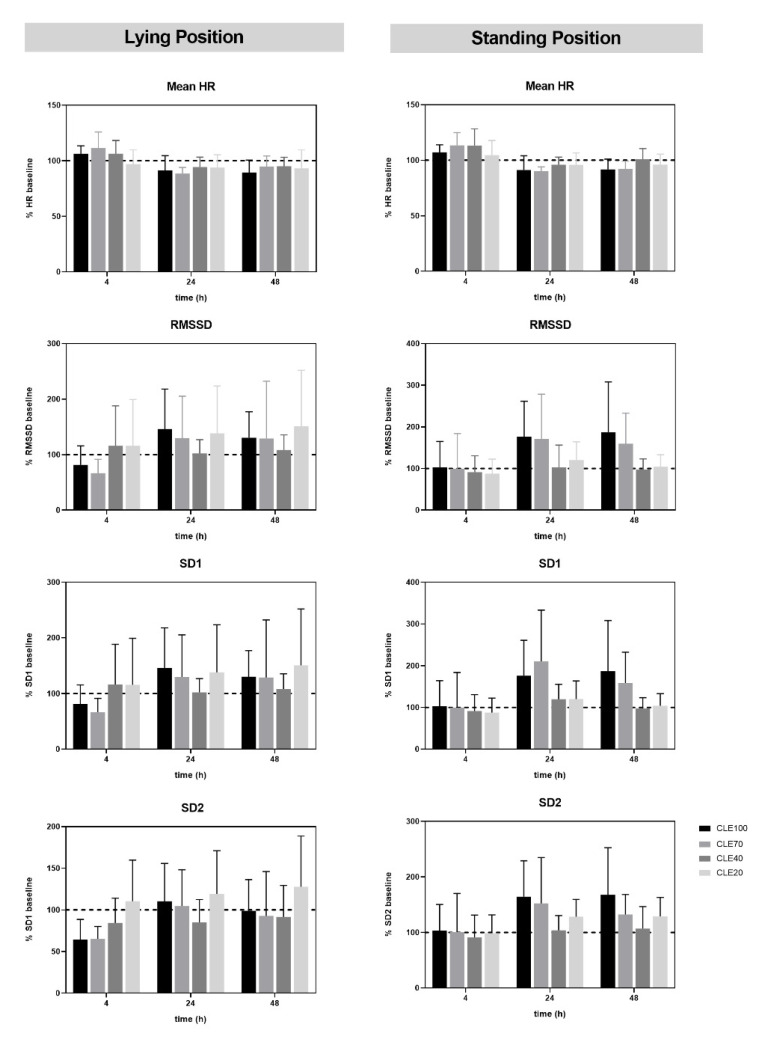
Percentage of baseline mean heart rate (HR) in lying and standing position, root mean square of successive RR interval differences (RMSSD), standard deviation of instantaneous beat-to-beat RR interval of Poincaré plot perpendicular to the line-of-identity (SD1) and along the line of identity (SD2) in lying and standing position measured 4, 24, and 48 h (Rec4, Rec24, Rec48) after four constant-load cycle ergometer exercise tests with maximum duration (CLE100) and 70, 40, and 20% of CLE100 (CLE70, CLE40, and CLE20) at the same intensity.

**Table 1 jfmk-07-00054-t001:** Heart rate (HR), oxygen uptake ($\dot{V}O_{2}$ ), lactate (La), and power output (P) at the first and second lactate (LTP_1_ and LTP_2_) and ventilatory thresholds (VT_1_ and VT_2_) as well as maximal values from the incremental cycle ergometer test.

	HR [bpm]	$\dot{V}O_{2}\;[L/\min]$	La [mmol/L]	P [W]
**LTP_1_**	118 ± 8	1.83 ± 0.37	1.17 ± 0.42	106 ± 24
**VT_1_**	118 ± 10	1.80 ± 0.46	1.19 ± 0.38	103 ± 0.71
**LTP_2_**	160 ± 10	3.26 ± 0.55	3.59 ± 1.00	212 ± 34
**VT_2_**	161 ± 10	3.26 ± 0.56	3.66 ± 1.14	213 ± 38
**Max**	187 ± 11	3.86 ± 0.75	12.31 ± 1.92	299 ± 48

**Table 2 jfmk-07-00054-t002:** Power output (P) at the first lactate turn point (LTP_1_) from the submaximal incremental cycle ergometer exercise test at baseline and 4, 24, and 48 h of recovery (Rec 4, Rec 24, and Rec 48) after each constant-load exercise test (CLE).

P LTP_1_	CLE20 (W)	CLE40 (W)	CLE70 (W)	CLE100 (W)
**baseline**	113 ± 16	107 ± 16	107 ± 18	106 ± 21
**Rec4**	112 ± 16	109 ± 17	108 ± 17	107 ± 19
**Rec24**	108 ± 17	108 ± 17	107 ± 17	106 ± 21
**Rec48**	112 ± 16	109 ± 19	108 ± 20	104 ± 19

## Data Availability

Not applicable.

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
