# Peer review of "Effects of Different Durations at Fixed Intensity Exercise on Internal Load and Recovery—A Feasibility Pilot Study on Duration as an Independent Variable for Exercise Prescription"

_jfmk, 2022, doi:10.3390/jfmk7030054_

Round 1

Reviewer 1 Report

Dear Authors

You have written an interesting paper. However, some parts need to be addressed for greater clarity.

From what I have read I would classify this as a pilot study. I would recommend adding this at the end of the title.

The introduction is well written and leads clearly to the study's rationale.

Methods - please identify/describe your sample more in detail and what you define under moderately trained.

How was the sample size determined? Was G*Power or any other method used? What were the inclusion and exclusion criteria? Report

What was the participant's physical activity level and what were their main exercise activities? Report

What was their experience in cycling activity? report

Why just 2 females? Elaborate

How were weight and height measured? report

Line 125 - state this local University.

Line 132 - name f and m abbreviations

Line 139 - Please report at what time of day were the tests performed

Line 152 - a few minutes? that could be 2 or 10? Be specific

What were the instructions to participants for physical activity between all tests and especially 1 day before measurements? What were your instructions for nutritional intake in these periods?

The limitations of the study paragraph could be extended. Could this be extended-applicable to/for running protocols, would performance be better if they would use cyclus 2 ergometer with their bike frames and settings?

Overall a well-written and straightforward study with well-developed discussion and with a solid foundation for further studies.

Therefore, I recommend acceptance after minor revisions.

Kind regards

Author Response

Dear Reviewer,

thank your for revising our manuscript. Please find below our replies to your comments.

Comment 1

From what I have read I would classify this as a pilot study. I would recommend adding this at the end of the title.

Reply 1.

We agree and included the term “pilot” in the title for clarity.

The introduction is well written and leads clearly to the study's rationale.

Comment 2

Methods - please identify/describe your sample more in detail and what you define under moderately trained.

Reply 2

Thank you for your comment. We addressed the mentioned issues in the method section: “subject characteristics”.

Comment 3

How was the sample size determined? Was G*Power or any other method used? What were the inclusion and exclusion criteria? Report

Reply 3

Information regarding the inclusion criteria was added (Line 128). As the study was a feasibility pilot-study we did not determine a sample size before the start of the study in order to gain data to calculate sample size retrospectively, which has been included in the text. (G-power calculation of sample size for the given study design).

Comment 4

What was the participant's physical activity level and what were their main exercise activities? Report

Reply 4

Revised, see Line 124 ff

Comment 5

What was their experience in cycling activity? Report

Reply 5

Revised, see Line 127 ff

Comment 6

Why just 2 females? Elaborate

Reply 6

We did not exclude females from the study, since in our opinion there was no physiological reason for this research question to do so. Unfortunately, only two female subjects were willing to participate in this rather strenuous study design. However, we agree that in future studies adequately powered groups of female and male subjects need to be investigated in order to detect sex differences.

Comment 7

How were weight and height measured? Report

Reply 7

Revised, see Line 171-173

Comment 8

Line 125 - state this local University. - revised

Comment 9

Line 132 - name f and m abbreviations

Reply 9

Abbreviations were already defined in the text before (Line 122)

Comment 10

Line 139 - Please report at what time of day were the tests performed

Reply 10

Thanks for this comment. All CLE tests were performed between 9 and 11 AM in order to organize the 4, 24 and 48 h recovery tests at regular times. Revised, see Line 145

Comment 11

Line 152 - a few minutes? that could be 2 or 10? Be specific -evised, see Line 158

Comment 12

What were the instructions to participants for physical activity between all tests and especially 1 day before measurements? What were your instructions for nutritional intake in these periods?

Reply 12

We addressed this issue in the Method section. Line 163 ff

We agree, however that in future fully powered studies it will be necessary to control nutrition and activity of subjects specifically after CLE tests.

Comment 13

The limitations of the study paragraph could be extended. Could this be extended-applicable to/for running protocols, would performance be better if they would use cyclus 2 ergometer with their bike frames and settings?

Reply 13

Thank you for this comment, we included some more limitations in the text. To our opinion the general principle will be the same, but due to the fact that running is a technical much more complex and a weight carrying sport, the durations and physiological parameters prescribing the internal load may be slightly different compared to cycling exercise. Using a cyclus 2 ergometer, may have some advantages I highly trained cyclists; however, this was not the intention of the study. We agree that including top level athletes, specificity is important and need to be respected in future studies.

Reviewer 2 Report

Dear authors,

congratulation for an article of interest, the results are limited by the number of subjects, but because it is the first time such study is done, it remain of interest for future readers, like a pilot study. May be you could explain why did you reach only 7 subjects. 

Wish of success in your publication 

Author Response

Dear Reviewer,

thank you for revising our manuscript and your wishes.

Our intention in this feasibility pilot-study was to proof the methodological approach. Due to the demanding study design, we kept the number of subjects small. However, the number of subjects analyzed is also small due to compliance and recruitment reasons, also related to the demanding study design.

Reviewer 3 Report

The Authors focused on an article of the Effects of duration at constant intensity exercise on internal load and recovery – a feasibility study on duration as an independent variable for exercise prescription.

The title does not clearly describe the article - I recommend a change.

The Abstract presents an accurate description of the case and its implications.

In the introduction, the author describes exactly what he wanted to accomplish and clearly defines the problem under study. I have to say that about 26% of references are from the last 5 years. Three figures and table in the text are very clearly written.

Overall impression about the quality of the study is good.

This is an interesting topic. In my opinion, too small a number of people taken into the study is a poor recommendation, while it provides a basis for undertaking further studies in a much broader group of patients.

A study is also needed among less-trained people with different ailments to confirm significant differences in physiological response. Are the authors considering this type of research?

Please note the superscript and subscript used in the article, please standardize throughout the paper.

Author Response

Dear Reviewer,

thank you for revising our manuscript. Please find below our replies to your comments.

Comment 1

The title does not clearly describe the article - I recommend a change.

Reply 1

Thanks for this comment, we slightly changed the title in order to increase clarity.

Comment 2

This is an interesting topic. In my opinion, too small a number of people taken into the study is a poor recommendation, while it provides a basis for undertaking further studies in a much broader group of patients.

Reply 2

Thanks for this comment. We agree that the number of subjects is small, but our intention in this feasibility pilot-study was to proof the methodological approach which has not been shown before. With this study we set the basis for fully powered future studies which need to include subjects with different performance, sex and sports specificity.

Comment 3

A study is also needed among less-trained people with different ailments to confirm significant differences in physiological response. Are the authors considering this type of research?

Reply 3

We fully agree regarding the need to investigate untrained subjects and even patients as the topic of optimal duration may be even more important to these groups of people such as mentioned by Mezzani et al. (2010, 2013) (see list of references). As these groups are rather vulnerable to high loads including unnecessary risks, we started the pilot with healthy trained subjects.

Comment 4

Please note the superscript and subscript used in the article, please standardize throughout the paper.

Reply 4

We checked and revised the text accordingly.

Reviewer 4 Report

Dear Authors,

as you correctly stated in your manuscript, due to the small sample size and to the missing control of some variables the work can be considered as a feasibility study of the methodology proposed. It sets a promising basis for follow-up studies to investigate the possibility to prescribe differences in the internal load and recovery kinetics with respect to duration at the same exercise intensity.

Some clarifications are needed and are listed point by points in the following comments.

Lines 50-52: in my opinion some further discussion on the topic of Intensity/Duration exercise prescription could be implement by the analysis of the following papers: https://doi.org/10.1123/ijspp.2020-0072; https://doi.org/10.3390/healthcare10061116.

Line 116: could you please explain the rational for including the orthostatic tests in your protocol?

Lines 135-136: for a clearer readability move “At least one week after the IE test, all subjects started with the all-out duration (CLE100) tests” to line 134, before “Workload for CE was set…”

Line 135: ascertain that all acronyms are defined the first time they are used, for example, Pmax and PLTP2.

Results section: is difficult to read, some sentences should be shortened or cut in two or rephrased, and some text could be better presented as tables.

Line 219: “Both P at LTP1 and VT1 as well as LTP2 219 and VT2 were not significantly different.” Clarify if different between them or something else.

Lines 248 and 250: what do you mean with “from the IE”? please clarify

Lines 263-267: “As one possibility to estimate the degree of fatigue for CLE70 and CLE100 without changes in external load, we suggested a calculated “virtual” performance loss. The difference in VEpeak gives an equivalent for power determined in the IE test which had to be compensated by means of an increase in VE from the mean value between 15 and 35 min of CLE to peak values at the end of CLE70 and CLE100 exercise”, the explanation is hard to follow, maybe it could be described in a graphical or mathematical form.

Lines 3552-353: “reducing intensity in small steps will additionally increase recovery time”, maybe you mean that the recovery time becomes shorter, please clarify.

Lines 369-370: “A post-hoc power analysis revealed about 100 subjects to prove significant effects for highly variable measures such as HRV”, if you are referring to a study with 100 subjects you must include the reference.

Lines 371-377: “Critically, it needs to be questioned if 4 h of rest are too long to analyse fatigue after different duration exercises, masked by rather fast recovery within a short period of time (within one h or even faster) [38–40]. Exercise tests directly after CLE, however, are rather demanding and motivation / activation potential of participants may be a severe limit for such an approach. From our perspective, the first recovery test should be closer to CLE termination and within one hour of recovery, a strategy which may be limited to trained subjects”, at this regard a previous work on normal weight non athletes females showed that, after different exercise modalities at 60% VO2peak, homeostasis is restored on average after 76 min of passive recovery, but already 60m minutes after exercise cessation VO2 is not statistically different form pre-exercise basal values (Galvani, C.; Bruseghini, P.; Annoni, I.; Demarie, S.; Salvati, A.; Faina, M. Excess Post-Exercise Oxygen Consumption after Different Moderate Physical Activities in a Healthy Female Population. Med. Sport 2013, 66, 2). Therefore, your suggestion that the first recovery test should be closer to CLE termination and within one hour of recovery, can be sustained. On the other hand, the suggestion that this strategy may be limited to trained subjects is not necessarily true. Indeed, your subjects presented a relative VO2peak around 53 ml/kg/min, while in the work I am suggesting you analyse subjects had a VO2peak around 40 ml/kg/min.

References: some references are really dated, in particular 3, 4 and 5, more recent citation could add value to your background, see for example Jamnick, NA.; Pettitt, RW.; Granata, C.; Pyne, DB.; Bishop, DJ. An Examination and Critique of Current Methods to Determine Exercise Intensity. Sports Med. 2020, 50, 10, 1729-1756 https://doi.org/10.1007/s40279-020-01322-8

Author Response

Dear Reviewer,

thank you for revising our manuscript. Please find below our replies to your comments.

Comment 1

Lines 50-52: in my opinion some further discussion on the topic of Intensity/Duration exercise prescription could be implement by the analysis of the following papers: https://doi.org/10.1123/ijspp.2020-0072; https://doi.org/10.3390/healthcare10061116.

Reply 1

Thank you for recommending these papers.

Regarding the Paper from Kesisoglou et al., we like the idea of determining a performance loss by the acute performance decrement (APD) following exercise with different intensities and also durations. We included this article in the introduction and discussion section

Regarding the paper from Gholizadeh et al. which is about the measurement of external load in football players, we understand the idea to include the monitoring and systematic measurement topic in our paper. As these approaches also contribute to a more individualised prescription of duration e.g. in terms of number of sprints. However, as our paper is about the general physiological effects of individualized duration, we think adding this paper would lead away from the specific duration topic, although worth mentioning.

Comment 2

Line 116: could you please explain the rational for including the orthostatic tests in your protocol?

Reply 2

Regarding to Hautala et a. (2009) (see list of references) the orthostatic test is a widely used method for quantifying autonomic nervous system activity in clinical practice, and its significance has been well-recognized in athletic training (Hedelin et al., 2001; Hynynen et al., 2007; Uusitalo et al., 2000). For that reason, we included this test in order to find a rather simple test version applicable in practice.

Comment 3

Lines 135-136: for a clearer readability move “At least one week after the IE test, all subjects started with the all-out duration (CLE100) tests” to line 134, before “Workload for CE was set…”

Reply 3

Good Point, thank you.

Comment 4

Line 135: ascertain that all acronyms are defined the first time they are used, for example, Pmax and PLTP2.

Reply 4

We controlled the text and revised the acronyms according to your recommendation.

Comment 5

Results section: is difficult to read, some sentences should be shortened or cut in two or rephrased, and some text could be better presented as tables.

Reply 5

We made some changes to the result section and added one table to increase readability.

Comment 6

Line 219: “Both P at LTP1 and VT1 as well as LTP2 219 and VT2 were not significantly different.” Clarify if different between them or something else.

Reply 6

We revised this accordingly. Line 235

Comment 7

Lines 248 and 250: what do you mean with “from the IE”? please clarify

Reply 7

Revised Line 265, 267

Comment 8

Lines 263-267: “As one possibility to estimate the degree of fatigue for CLE70 and CLE100 without changes in external load, we suggested a calculated “virtual” performance loss. The difference in VEpeak gives an equivalent for power determined in the IE test which had to be compensated by means of an increase in VE from the mean value between 15 and 35 min of CLE to peak values at the end of CLE70 and CLE100 exercise”, the explanation is hard to follow, maybe it could be described in a graphical or mathematical form.

Reply 8

We agree and add an example of a calculation for better understanding. (Line 287 ff)

Comment 9

Lines 3552-353: “reducing intensity in small steps will additionally increase recovery time”, maybe you mean that the recovery time becomes shorter, please clarify.

Reply 9

According to the concept of Platonov, strongest fatigue and biggest adaptive effects are suggested after exercise with a duration exceeding the point of compensated fatigue. In our study we did not exceed this point. However, when exceeding this point, this is suggested to cause stronger effects and longer recovery time. In a laboratory setting you practically would have to reduce the performance in small steps to bring the subject to the point of “total” fatigue.

Comment 10

Lines 369-370: “A post-hoc power analysis revealed about 100 subjects to prove significant effects for highly variable measures such as HRV”, if you are referring to a study with 100 subjects you must include the reference.

Reply10

In this sentence we are not referring to a study. Based on our data we performed a power analysis to estimate the necessary number of subjects to get significant results with regard to further studies with the given complex study design.

Comment 11

Lines 371-377: “Critically, it needs to be questioned if 4 h of rest are too long to analyse fatigue after different duration exercises, masked by rather fast recovery within a short period of time (within one h or even faster) [38–40]. Exercise tests directly after CLE, however, are rather demanding and motivation / activation potential of participants may be a severe limit for such an approach. From our perspective, the first recovery test should be closer to CLE termination and within one hour of recovery, a strategy which may be limited to trained subjects”,

at this regard a previous work on normal weight non athletes females showed that, after different exercise modalities at 60% VO2peak, homeostasis is restored on average after 76 min of passive recovery, but already 60m minutes after exercise cessation VO2 is not statistically different form pre-exercise basal values (Galvani, C.; Bruseghini, P.; Annoni, I.; Demarie, S.; Salvati, A.; Faina, M. Excess Post-Exercise Oxygen Consumption after Different Moderate Physical Activities in a Healthy Female Population. Med. Sport 2013, 66, 2). Therefore, your suggestion that the first recovery test should be closer to CLE termination and within one hour of recovery, can be sustained. On the other hand, the suggestion that this strategy may be limited to trained subjects is not necessarily true. Indeed, your subjects presented a relative VO2peak around 53 ml/kg/min, while in the work I am suggesting you analyse subjects had a VO2peak around 40 ml/kg/min.

Reply 11

Thank you for your detailed analysis and constructive comments. Based on our experience and with regard to future studies, their feasibility, quality (i.a. of multiple all out tests) and compliance of subjects we argued the limitation to trained subjects. However, we agree that this might not be limited to trained subjects and changed the term to “subjects experienced in exercise training”:

Comment 12

References: some references are really dated, in particular 3, 4 and 5, more recent citation could add value to your background, see for example Jamnick, NA.; Pettitt, RW.; Granata, C.; Pyne, DB.; Bishop, DJ. An Examination and Critique of Current Methods to Determine Exercise Intensity. Sports Med. 2020, 50, 10, 1729-1756 https://doi.org/10.1007/s40279-020-01322-8

Reply 12

Thank you for your literature suggestion. We replaced two citations in the introduction (Line 34-36) with the actual paper from Jamnick. However, we must keep the references 3,4, and 5 as our manuscript is built on the theoretical basics of these references.